# Post-COVID-19 Syndrome Based on Disease Form and Associated Comorbidities

**DOI:** 10.3390/diagnostics12102502

**Published:** 2022-10-15

**Authors:** Ramona Cioboata, Dragos Nicolosu, Costin Teodor Streba, Corina Maria Vasile, Madalina Olteanu, Alexandra Nemes, Andreea Gheorghe, Cristina Calarasu, Adina Andreea Turcu

**Affiliations:** 1Department of Pneumology, University of Pharmacy and Medicine Craiova, 200349 Craiova, Romania; 2Pneumology Department, Victor Babes University Hospital Craiova, 200515 Craiova, Romania; 3Department of Pediatric Cardiology, “Marie Curie” Emergency Children’s Hospital, 041451 Bucharest, Romania; 4Faculty of Dentistry, University of Pharmacy and Medicine Craiova, 200349 Craiova, Romania; 5Memorial Life Hospital Bucharest, 010719 București, Romania; 6PhD School Department, University of Pharmacy and Medicine Craiova, 200349 Craiova, Romania; 7Infectious Disease Department, Victor Babes University Hospital Craiova, 200515 Craiova, Romania

**Keywords:** COVID-19, pneumonia, long COVID syndrome, associated comorbidities, SARS-CoV-2

## Abstract

(1) Background: SARS-CoV-2 has infected more than 97 million people worldwide and caused the death of more than 6 million. (2) Methods: Between 1 October and 31 December 2020, 764 patients diagnosed with SARS-CoV-2 infection were selected based on RT-PCR test results. The following parameters were noted: age, gender, origin, days of hospitalization, COVID-19 experienced form, radiographic imaging features, associated comorbidities, and recommended treatment at discharge. (3) Results: The mean age at the time of COVID-19 infection was 55.2 years for men and 55.3 years for women. There was a similar age distribution among patients, regardless of gender. There was a substantial difference between the average lengths of hospitalization and those with residual symptoms—most patients who reported symptoms after discharge had been admitted with moderately severe forms of illness. Fatigue was the main remaining symptom (36%). (4) Conclusions: In conclusion, to clarify the impact of SARS-CoV-2 infection on patients in the long term, further studies are needed to investigate the elements assessed. Well-designed recovery programs will be needed to effectively manage these patients, with multidisciplinary collaboration and a team of professionals involved in all aspects of post-COVID patient health.

## 1. Introduction

Coronavirus disease 2019 (COVID-19) is a highly contagious condition of the respiratory system caused by the novel coronavirus called severe acute respiratory syndrome coronavirus 2 (SARS-CoV-2) [1]. SARS-CoV-2 has thus far infected more than 97 million people worldwide and caused the death of more than 6 million [2]. The symptoms caused may include asymptomatic cases, fever, fatigue, muscle pain (myalgia), mild upper airway infection, and severe life-threatening pneumonia, and can lead to death, especially in the elderly and those with comorbidities [3].

COVID syndrome is characterized by symptoms and signs related to severe acute respiratory syndrome coronavirus 2 (SARS-CoV-2) present at least four weeks after acute infection [4]. Furthermore, it can be described as either continuous symptomatic COVID-19 from 4 to 12 weeks or post-COVID-19 syndrome from 12 weeks onwards [1]. Long-standing COVID-19 symptoms and signs are weakly described and may be associated with significant morbidity [5].

Post-COVID symptoms usually occur among people who have experienced a mild form and those with severe and hospitalized episodes. Categories at increased risk are adults over 50, people diagnosed with a severe form of COVID, with pre-existing conditions, especially cardiovascular, hypertension, diabetes, or obesity [6].

## 2. Materials and Methods

### 2.1. Aim of the Study

This retrospective, observational, descriptive study with an analytical component was performed to identify post-COVID-19 symptoms among a sample of 764 patients from the only COVID support hospital in the Oltenia region of Romania. This study aimed to highlight persistent symptoms within six months after COVID-19 infection according to the form of disease suffered, the number of days of hospitalization required, and other associated comorbidities.

### 2.2. Study Design and Population

The cohort group of this research enrolled 764 patients diagnosed with SARS-CoV-2 infection by RT-PCR between 1 October and 31 December 2020, at the Hospital of Infectious Diseases and Pneumophthiology in Craiova, Romania.

The following parameters were noted: age, gender, origin, days of hospitalization, experienced form COVID-19 (mild, moderate, or severe), imaging features on radiography (marked interstitial, pneumonia appearance, or bronchopneumonia pattern), associated comorbidities (diabetes mellitus, hypertension, obesity, chronic kidney disease, liver disease, and others) and recommended treatment at discharge.

Patients were contacted by telephone between 15 April and 15 May 2021 and questioned about the existence and duration of the main symptoms experienced after discharge: fatigue, myalgia, dyspnea, headache, cough, and neuropsychological disorders.

Participants were selected according to the inclusion and exclusion criteria. Inclusion criteria in our study: age over 18 years, positive PCR test, and patients who signed the study inclusion consent. Exclusion criteria in our study: pregnant women, patients with psychiatric disorders, and patients who did not provide sufficient data on remaining symptoms after SARS-CoV-2 infection.

We categorized disease severity in accordance with the World Health Organization (WHO) COVID-19 disease severity classification: mild disease (symptomatic patients confirmed with SARS-CoV-2 infection but without signs of viral pneumonia or hypoxia), moderate disease (patients with clinical signs of pneumonia but no signs of severe pneumonia on chest X-ray and SpO2 ≥ 90% on room air), severe disease (patients with clinical and radiological signs of severe pneumonia plus SpO2 < 90% on room air or respiratory rate > 30 breaths/min) [7].

Patients included in the study received treatment according to the form of the disease, as follows: those with mild forms of the disease received symptomatic medication (antipyretics, antipyretics, nasal decongestants, anti-inflammatories), patients with moderate forms of the disease received antivirals (Remdesivir), and those with severe forms received antivirals and corticosteroids.

### 2.3. Ethical Principles

In this study, written informed consent was obtained from all patients regarding the processing of data for scientific purposes. The study was conducted in accordance with the guidelines of the Declaration of Helsinki and approved by the Local Ethics Committee (approval no. 11346/15.05.2021).

### 2.4. Statistical Analysis

In this study, the distribution of cases was analyzed according to the form of the disease (mild, moderate or severe), the sex of the patient, other associated pathologies in relation to the number of days of hospitalization required, and the environment of origin.

Results were expressed descriptively as the number of cases, prevalence (%), and percentage changes for the categories analyzed. All patient information was tabulated in an electronic database using GraphPad Prism Demo 6.0 software (GraphPad, San Diego, CA, USA) for data analysis, parallel with the Microsoft Excel statistics software suite (Microsoft, Albuquerque, NM, USA).

Data descriptive statistics and histograms were derived to observe the distribution of the data, using Student’s *t*-test, chi-square, and Pearson correlation coefficient as appropriate for analysis.

## 3. Results

Our study’s cohort group included 764 patients who completed follow-ups within six months of testing positive for COVID-19. Of the patients included in the study, 48.03% were women, and the mean age at the time of infection with COVID-19 was 55.2 years for men and 55.3 years for women (Figure 1). There was a similar age distribution among patients regardless of sex (F-test for comparison of variances, *p* = 0.764).

As for the main comorbidities associated with COVID-19, patients included in the study group had hypertension (36%). In second place was type 2 diabetes mellitus, observed in 14.52%. Additionally, 29 of the study group were obese (3.8% of the group were also associated with diabetes mellitus). Chronic kidney disease was reported in 18 patients (2.35%) and chronic liver disease in 12 patients (1.6%).

Among the 764 patients, only 1 in 16 (4.35%) women and 17 (4.28%) men required more than 25 days of hospitalization (Figure 2).

Most patients (63.3% of the cohort) were classified with moderate disease (261 men and 222 women, 65.7% and 60.5%, respectively) (Figure 3).

There were no significant differences between the two sexes or backgrounds in the prevalence of this clinical form (Figure 4).

There were 234 patients (30.7%) in the study group with mild forms, but no significant differences could be detected between the sexes or by background.

There were no significant differences between the two sexes or the environment in the proportion of severe forms (chi-square test, male/female *p* = 0.344; urban/rural *p* = 0.119). Among them, we identified 46 people (6%) with severe forms of the disease.

The study reported 24 known patients with chronic bronchoconstrictive disease (16 asthmatics, 8 with COPD) and five with sequelae stroke. Four post-discharge deaths were in patients aged 60–74 years, two from urban and two from rural areas, with lengths of hospital stay between 8 and 22 days. All the deceased had been admitted with severe forms of disease and associated comorbidities—two people with diabetes, one with old myocardial infarction, and one patient with obstructive sleep apnea syndrome and ischemic stroke. Regarding the main comorbidities encountered in patients with COVID-19 disease included in the study group, most had hypertension (36%). Diabetes mellitus type 2 was the other main comorbidity (14.52%).

Additionally, 29 of the study group were obese (3.8% of the group were also associated with diabetes). Chronic kidney disease was recorded in 18 patients (2.35%) and chronic liver disease in 12 patients (1.6%).

A solid correlation was noted between patients’ ages and the number of days of hospitalization, no matter their sex or background (Pearson’s r correlation coefficient, *p* < 0.0001). Therefore, older ages require more extended hospitalization, even with moderate or mild forms of the disease, mainly because of the persistence of specific symptomatology and more delayed negativity (Table 1).

Regarding radiographic findings, bilateral patterns prevailed (476 patients, 62.3%), with a marked interstitial appearance, specific to the condition, reported in 216 patients with bilateral lesions and 208 with unilateral lesions (424 patients, 55.5%).

The pneumonic form was observed in 233 patients (30.5%), while the bronchopneumonic form affected 100 patients (13%) (Table 2).

There was a strong correlation between the pneumonic forms and the increased severity of the disease (100% of severe forms had the appearance of bilateral pneumonia or bronchopneumonia, *p* < 0.0001). However, 214 patients with mild forms of the disease (91.4% of all mild forms, 28% of the group) reported marked uni- or bilateral interstitial changes in their radiology examination (Pearson correlation coefficient r 0.9 *p* < 0.0001) (Table 3).

There were 352 patients (46%) in the cohort who reported various remaining post-discharge symptoms, with mean durations between 19 and 26 days (median 16 days, minimum 2, maximum 30 days). A significant difference was observed between the mean lengths of hospitalization and those with remaining symptoms (Student’s *t*-test, *p* < 0.0001). Most patients who developed symptoms after discharge were admitted with moderate forms of the disease. Fatigue was the main remaining symptom (275 reports), but the most persistent was cough (26 days on average), followed by myalgia (23 days on average).

The neuropsychiatric manifestations observed among the patients enrolled in our study are described in the table below (Table 4).

## 4. Discussion

Bin Cao, affiliated with the National Center for Respiratory Medicine and co-author of a study published in *The Lancet*, 2021, that tracked the long-term manifestations of SARS-CoV-2 infection, said, “Because COVID-19 is such a new disease, we are only now beginning to understand some of the long-term health effects on patients. Our work highlights the importance of conducting longer-term follow-up studies in larger populations to understand the full spectrum of effects that COVID-19 may have on humans” [4].

The study published by Huang et al. [5] included 1733 COVID-19 patients discharged between January and May 2020 from Jin Yin-tan Hospital, Wuhan City, China. The mean age of the study participants was 57 years. In our study, the average age of study participants was 55 years.

In our study, most patients who experienced symptoms after discharge had previously been admitted with moderate forms of the disease. Fatigue was the main remaining symptom (36%), but the most persistent was cough (on average of 26 days), followed by myalgia (on average of 23 days). Among the study group, 352 patients (46%) reported various remaining post-discharge symptoms, with mean durations between 19 and 26 days (median 16 days, minimum 2, maximum 30 days). We observed a significant difference between mean lengths of hospitalization and those with remaining symptoms (Student’s *t*-test, *p* < 0.0001).

In a study by Huang et al. [5], six months after COVID-19 infection, 76% of patients had at least one symptom. Of the study participants, 63% had fatigue or muscle weakness, 26% had sleep disturbances, and 23% reported anxiety or depression. Patients with severe forms of COVID-19 had more severe lung diffusion impairment and lung imaging abnormalities. In 13% of study participants with normal kidney function during hospitalization, laboratory tests showed decreased kidney function at follow-up evaluation.

Another Italian study [6] reported the persistence of symptoms in 87.4% of 143 patients discharged from the hospital after acute COVID-19, followed by 60 days after the first symptom onset. Patients more commonly experienced: fatigue (53.1%), dyspnea (43.4%), joint pain (27.3%), chest pain (21.7%), 55% of patients continued to have three or more symptoms, and a decrease in quality of life, as measured by the EuroQoL visual analog scale, was observed in 44.1% of patients. Similar results were also reported in a French study [8] of 150 COVID-19 non-critical patients after discharge, 60 days after follow-up, and reported persistence of symptoms in 66% of the cohort. Other prospective studies and surveys have reported similar results. Fatigue, dyspnea, and psychological disorders (post-traumatic stress disorder, anxiety, depression, and problems with concentration and sleep) were reported by >30% of groups during the 8–14 weeks of follow-up after discharge [9,10,11,12].

According to the retrospective study by Osikomaiya et al. [13], among 274 patients who attended COVID-19 outpatient clinics in Lagos State, more than one-third (41%) had persistent COVID-19 after discharge, and approximately 20% had more than three persistent COVID-like symptoms. The most persistent COVID-19-like symptoms reported were mild headache (13%), fatigue (13%), and chest pain (10%).

Furthermore, in a previous study of 438 COVID-19 survivors in Wuhan, China, three months or more after hospital discharge, they were assessed for post-COVID symptoms [14]. Persistent symptoms were shared, including general symptoms (50%), respiratory symptoms (39%), cardiovascular-related symptoms (13%), psychosocial symptoms (22.7%), and alopecia (28.6%).

Cardiac magnetic resonance imaging (MRI) suggests that persistent myocardial inflammation may be present in up to 60% of patients two months after COVID symptoms. However, the reproducibility and consistency of these data have been debated [15].

Evidence from a study of 26 competitive athletes diagnosed with mild or asymptomatic SARS-CoV-2 infection, cardiac MRI revealed features of myocarditis in 15% of participants and previous myocardial damage in 30.8% of subjects [16].

Retrospective data on post-acute thromboembolic events suggest venous thromboembolism (VTE) rates < 5% in the post-acute COVID-19 period [17]. The risk of thrombotic complications in the post-acute COVID-19 phase is likely related to the duration and severity of a hyper-inflammatory state. It is unknown how long it persists. Severe acute kidney injury (AKI) requiring dialysis occurs in 5% of all hospitalized patients and 20–31% of critically ill patients with acute COVID-19, especially those mechanically ventilated [18,19]. No significant gastrointestinal and hepatobiliary sequelae have been reported in COVID-19 survivors [20,21].

In COVID-19, severe forms, as in other critical illnesses, cause decreased muscle mass through increased catabolism and feeding difficulties. Malnutrition was reported in 26–45% of patients with COVID-19 in an Italian study [22].

Protocols for nutritional support are mainly established for patients with severe respiratory distress, nausea, diarrhea, and anorexia with reduced food intake [23,24].

It seems that infection with the novel coronavirus also affects the mental state, not just the body. According to previously published studies, one in five people diagnosed with COVID-19 reported anxiety disorders, depression, or insomnia within three months of infection [25,26]. However, only 8.5% of our patients experienced neuropsychiatric disorders six months after SARS-CoV-2 infection.

In our study, only 3.8% of patients were associated with obesity. Patients who had obesity were also diagnosed with diabetes. This has also been reported in other studies [27] which demonstrated that obesity is associated with poor outcomes after SARS-CoV-2 infection. The evidence suggests that not only obesity but also overweight (i.e., a BMI of >25 kg/m^2^) is linked to more severe forms of the infection. In the study by Gao et al. [28], the risk ratio of severe consequences of COVID-19 progressively increased above a BMI of 23 kg/m^2^.

Statistically, many people will have at least one depressive episode following infection with COVID-19. According to a study by Huang C. et al. [29], depression can be diagnosed with a 2% probability within 14 to 90 days of infection with the new coronavirus.

## 5. Limitations

There are several strengths and limitations to this study. To the best of our knowledge, this was the first study focusing on the long-term effects of COVID-19 conducted in Romania, with a significant impact on participants. Additional studies may be needed to investigate the role of comorbidities in the development of long COVID syndrome. This study has several limitations. It was conducted in a single center; therefore, it was based on patient-reported symptoms without any objective assessment, which implies a distortion of information. There has only been one PCR test performed. The follow-up period was not homogeneous for participants, thus not allowing a division of patients into follow-up groups based on time intervals.

## 6. Conclusions

In conclusion, a prospective follow-up of a large cohort of recovered COVID-19 patients of varying severity with different backgrounds suggested that nearly 50% of them had post-COVID-19 syndrome six months after onset. Several clinical and healthcare management issues should be considered in a long-term pandemic. In addition, the association of post-COVID-19 symptoms with serological response triggers more insights into the underlying pathogenetic mechanisms and possible therapeutic strategies.

Therefore, to clarify the impact of SARS-CoV-2 infection on patients in the long term, further studies are needed to further investigate the elements assessed.

Post-acute COVID-19 (“long COVID”) is a multisystem disease with a complex long-term impact on patients that is not fully documented.

There will be a need for well-designed recovery programs to effectively manage these patients, with multidisciplinary collaboration and a team of professionals involved in all aspects of post-COVID patient health.

## Figures and Tables

**Figure 1 diagnostics-12-02502-f001:**
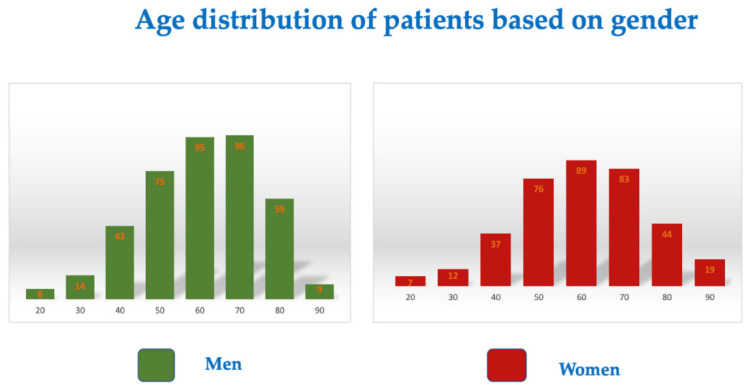
Age distribution based on gender.

**Figure 2 diagnostics-12-02502-f002:**
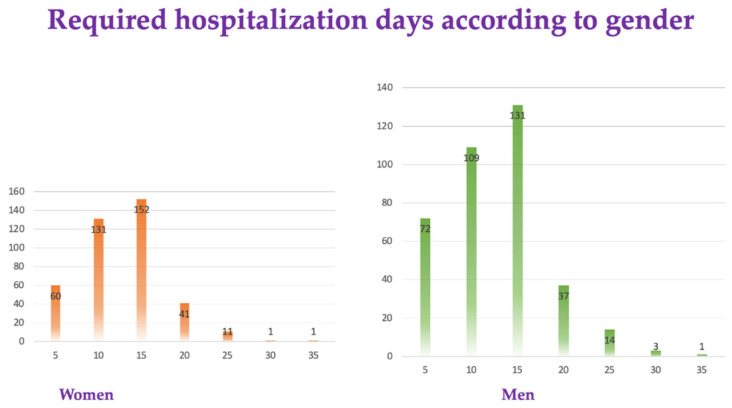
Required hospitalization days according to gender.

**Figure 3 diagnostics-12-02502-f003:**
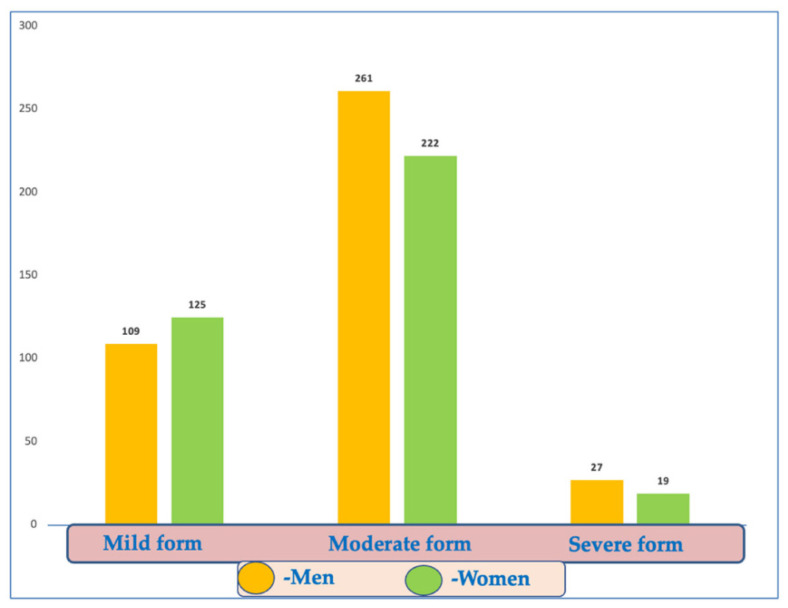
Distribution of COVID-19 disease forms according to gender.

**Figure 4 diagnostics-12-02502-f004:**
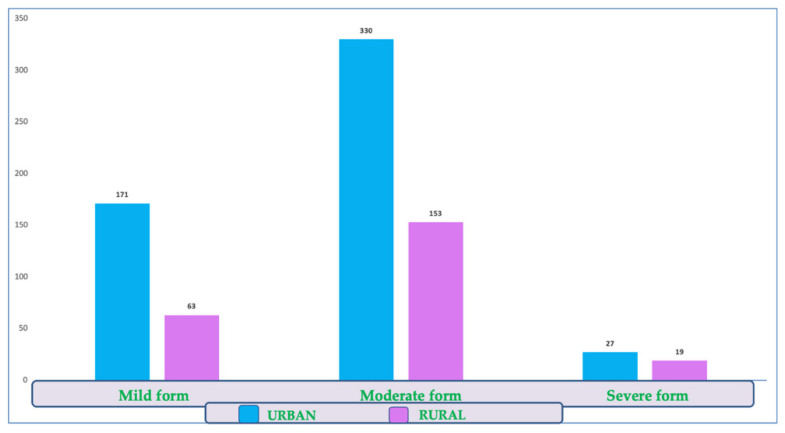
Distribution of COVID-19 disease forms by environmental background.

**Table 1 diagnostics-12-02502-t001:** Pearson’s r correlation coefficient according to age and hospitalization duration.

Pearson r (Based on Age and Days of Hospitalization)	Female	Male
Correlation coefficient r	0.50	0.35
Confidence interval 95%	0.42–0.57	0.27–0.44
P (two-tailed)	<0.0001	<0.0001

**Table 2 diagnostics-12-02502-t002:** Persistent symptoms after SARS-CoV-2 infection in relation to associated comorbidities, the form of disease experienced, and days of hospitalization required.

Symptom	No	%	Average Days	Classification							
				Mild	Moderate	Severe	Diabetes	Hypertension	Obesity	Pneumonic Form	Hospitalization Days >15
Fatigability	275	36.1%	22.4	94	181	13	32	104	14	121	44
Myalgia	75	9.8%	23.1	17	51	7	12	22	1	34	12
Dyspnea	79	10.3%	21	15	60	4	14	32	1	37	10
Headeache	71	9.3%	19	23	42	6	10	24	4	29	11
Cough	62	8.1%	26.4	22	37	3	5	19	1	23	9
Neuropsychiatric disorders	67	8.5%	N/A	19	46	2	8	27	4	30	12

**Table 3 diagnostics-12-02502-t003:** Student’s *t*-test (symptomatology duration compared with age, hospitalization duration, disease form, radiological form and diabetes mellitus).

Symptomatology Duration Compared With	Age	Hospitalization Duration	Disease Form	Radiological Form	Diabetes Mellitus
Pearson’s r Coefficient	0.008557	−0.06014	−0.01386	−0.03059	0.005889
95% Confidence interval	−0.11–0.12	−0.18–0.058	−0.13–0.10	−0.15–0.09	−0.11–0.12
P (two-tailed)	0.88	0.32	0.82	0.61	0.92

**Table 4 diagnostics-12-02502-t004:** Neuropsychiatric manifestations.

Anxiety	Adjustment disorders	This occurred no more than 3 months after the onset of the triggering factor. The intensity of the manifestations can be very intense and affect the person’s social and professional life.
	Anxiety disorder	Typically characterized by excessive worry about all aspects of life.
	Panic disorder	A state of intense fear or discomfort suddenly experienced. Symptoms of a panic attack are palpitations or rapid heartbeat, chest pain, shaking, shortness of breath, feeling of drowning, dizziness or fainting, sweating, numbness or tingling, fear of losing control or “going crazy”, fear of death.
Depressive episode	Defined as a period of acute sadness that persists for at least two weeks, expressed mainly by:	
	-Poor mood and/or loss of interest in most activities;	
	-Marked fatigue;	
	-Insomnia or hypersomnia;	
	-Feelings of worthlessness and helplessness;	
	-Changes in appetite or libido;	
	-Recurrent thoughts of death.	
Severe insomnia	Recognizable by symptoms such as:	
	-Inability to fall asleep for fear of something happening to them;	
	-Fear of death;	
	-Worry and fatigue.	
	Insufficient or poor sleep quality can have serious consequences: fatigue, concentration disorders, irritability, irritability, and low energy.	

## Data Availability

Not applicable.

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
