# Peer review of "Post-COVID-19 Syndrome Based on Disease Form and Associated Comorbidities"

_diagnostics, 2022, doi:10.3390/diagnostics12102502_

Round 1

Reviewer 1 Report

The manuscript describes the post COVID-19 syndrome observed in a cohort of 764 patients from hospital in Romania. The Authors examined the relationship of its symptoms with the presentation of COVID-19, the number of days of hospitalization, the gender and age of the patients and their comorbidities, and other factors.

I have some specific comments:

1. Abstract is too long. According to the Instructions for Authors, it should contain a maximum of 200 words. Please rephrase the Abstract.

2. Abstract: please correct typos, e.g., line 28 (should be COVID-19, without space), line 30 (remove the bracket), line 36 (remove dot).

3. Introduction: line 45, first provide the full name of COVID-19, and in brackets the abbreviation.

4. Introduction: line 46, comment as above, concerns SARS-CoV-2.

5. Introduction: line 53, please use only the abbreviation, concerns SARS-CoV-2.

6. Materials and Methods: scientific publications usually do not use personal forms like "we". Please rephrase this part of manuscript (e.g., lines 64, 71, 90).

7. Materials and Methods: line 88, please remove one bracket.

8. Results: comment the same as point 6., concerns “we”, “our”, etc. Please rephrase this part of manuscript.

9. Results: line 99, epidemiological papers describing patients use “study groups of patients” or “cohort”.  The phrase "study included 764 patients" is unfortunate. Please correct it in all manuscript.

10. Results: Figures 1 and 2 are illegible. Please improve their resolution or present the data in a table.

11. Results: please combine Tables 1 and 2.

12. Results: Table 3 and 4 were not prepared according to the Instructions for Authors. Please correct them.

13. Discussion: lines 250-279, these are the results. Please put them in the Results in the form of a table.

Author Response

Dear Reviewer,

Thank you very much for your review and constructive suggestions.

We have made extensive editing of the paper (both the information and the page display of the text) to improve its quality and to make it easier to follow. We have rewritten some paragraphs in order to convey the information more clearly, and we have also eliminated repeating information.

The manuscript describes the post COVID-19 syndrome observed in a cohort of 764 patients from hospital in Romania. The Authors examined the relationship of its symptoms with the presentation of COVID-19, the number of days of hospitalization, the gender and age of the patients and their comorbidities, and other factors. 

I have some specific comments:

Q1. Abstract is too long. According to the Instructions for Authors, it should contain a maximum of 200 words. Please rephrase the Abstract.

A1: We have revised the abstract and modified it accordingly.

  Q2. Abstract: please correct typos, e.g., line 28 (should be COVID-19, without space), line 30 (remove the bracket), line 36 (remove dot).

A2 : We have corrected the typos . Thank you for pointing that out.

Q3. Introduction: line 45, first provide the full name of COVID-19, and in brackets the abbreviation.

A3 : We modified it accordingly.

Q4. Introduction: line 46, comment as above, concerns SARS-CoV-2.

A4: We modified it accordingly.

Q5. Introduction: line 53, please use only the abbreviation, concerns SARS-CoV-2.

A5: We modified it accordingly.

Q6. Materials and Methods: scientific publications usually do not use personal forms like "we". Please rephrase this part of manuscript (e.g., lines 64, 71, 90).

A6: We have rephrased this part of the manuscript accordingly.

Q7. Materials and Methods: line 88, please remove one bracket.

A7: We modified it accordingly.

Q8. Results: comment the same as point 6., concerns “we”, “our”, etc. Please rephrase this part of manuscript.

A8: We rephrased this part of the manuscript accordingly.

Q9. Results: line 99, epidemiological papers describing patients use “study groups of patients” or “cohort”.  The phrase "study included 764 patients" is unfortunate. Please correct it in all manuscript.

A9: We modified it accordingly and rephrased the whole manuscript.

Q10. Results: Figures 1 and 2 are illegible. Please improve their resolution or present the data in a table.

A10: We tried to improve the quality of the pictures in the manuscript.

Q11. Results: please combine Tables 1 and 2.

A11: Following your suggestion, we combined Tables 1 and 2. Thank you!

Q12. Results: Table 3 and 4 were not prepared according to the Instructions for Authors. Please correct them.

Q12: We corrected Tables 3 and 4 accordingly.

Q13. Discussion: lines 250-279, these are the results. Please put them in the Results in the form of a table.

A13: We moved this paragraph to the Results sections in a form of a table.

Reviewer 2 Report

Dear colleagues,

The following items should be corrected:

-          The keywords should be more than five.

-          Figure 1, 2, 3 and 4 are not clear, it should be presented in one table with statistic analysis data and other characteristics of patients.  It is not results of the study. This data should be presented in the part Materials and Methods.

-          Table 1 and 2 are not clear.

-          It is recommended to structure the article, presenting the aim of the study, characteristic of patients in the materials and methods with a description of the design of the study.

-          Colleagues should include the definition of mild, moderate, or severe COVID-19 with inclusion in references WHO recommendations In the design of the study.

-          It is necessary to analyze correlation between a therapy that patients have been used in COVID-19 period and post-COVID-19 characteristics.

-          The inclusion and exclusion criteria were not presented clear in the Materials and Methods of the article.

-          Table 3 should be corrected with statistic data and the study design.

-          Conclusion is not clear. Colleagues did not provide statistically significant evidence of conclusion data.

-          References should be corrected. Some of references are not clear (8, 9, 10, 11, 14, 15, 18,19,20 and other…)

All these concerns should be well addressed to consider this manuscript suitable for publication.

Author Response

Dear Reviewer,

Thank you very much for your review and constructive suggestions.

We have made extensive editing of the paper (both the information and the page display of the text) to improve its quality and to make it easier to follow. We have re-written some paragraphs in order to convey the information more clearly, and we have also eliminated repeating information.

Dear colleagues,

The following items should be corrected:

Q1: The keywords should be more than five. 

A1: Thank you for pointing this out. We modified it accordingly.

Q2:   Figure 1, 2, 3 and 4 are not clear, it should be presented in one table with statistic analysis data and other characteristics of patients.  It is not results of the study. This data should be presented in the part Materials and Methods. 

A2:  We tried to improve the quality of the images. We decided to keep the images and not turn them into tables. We believe that the charts we use provide a good overview for readers.

Q3: Table 1 and 2 are not clear. 

A3:  We improved Tables 1 and 2.

Q4:  It is recommended to structure the article, presenting the aim of the study, characteristic of patients in the materials and methods with a description of the design of the study.

A4: Thank you for pointing this. We modified the design of our study and added essential data that was missing.

Q5:  Colleagues should include the definition of mild, moderate, or severe COVID-19 with inclusion in references WHO recommendations In the design of the study. 

A5: At your suggestion, we added the classification of COVID-19 disease forms according to WHO guidelines.

For the assessment of diagnostic severity, we used chest X-ray images, respiratory function parameters (oxygen saturation) and hematological parameters according to the World Health Organization (WHO) COVID-19 disease severity: mild disease (symptomatic patients confirmed with SARS-CoV-2 infection but without signs of viral pneumonia or hypoxia), moderate disease (patients with clinical signs of pneumonia but no signs of severe pneumonia on chest X-ray and SpO2 ≥ 90% on room air), severe disease (patients with clinical and radiological signs of severe pneumonia plus SpO2 < 90% on room air or respiratory rate >30 breaths/min) [20].

 Ref: Clinical management of COVID-19: Interim guidance: WHO/2019-nCoV/clinical/2021.1. Available online: https://apps.who.int/iris/handle/10665/332196 (accessed on 10 October 2022).

Q6:  It is necessary to analyze correlation between a therapy that patients have been used in COVID-19 period and post-COVID-19 characteristics. 

A6: Following your suggestion, we added information regarding the treatment received by the patients included in our study. The possible side effects of the medication received in COVID-19 are not the subject of our study, therefore, we have not included data on this. Thank you.

Patients included in the study received treatment according to the form of the disease, as follows: those with mild forms of the disease received symptomatic medication (antipyretics, antipyretics, nasal decongestants, anti-inflammatories), patients with moderate forms of the disease received antivirals (Remdesivir), and those with severe forms received antivirals and corticosteroids.

Q7:  The inclusion and exclusion criteria were not presented clear in the Materials and Methods of the article. 

A7: We appreciate the suggestion. We added inclusion and exclusion criteria to our study.

Inclusion criteria in our study: age over 18 years, PCR positive test, and patients who have signed the study inclusion consent.

Exclusion criteria in our study: pregnant women, patients with psychiatric disorders, and patients who did not provide sufficient data on remaining symptoms after SARS-CoV2 infection.

Q8:     Table 3 should be corrected with statistic data and the study design.

A8:  We modified the design of Table 3 : Persistent symptoms after SARS-CoV-2 infection in relation to associated comorbidities, the form of disease experienced, and days of hospitalization required.

Q9:  Conclusion is not clear. Colleagues did not provide statistically significant evidence of conclusion data. 

This study has several limitations. As it was conducted in a single center, it was based on patient-reported symptoms without any objective assessment, which implies a distortion of information. There has been only one PCR test performed.

In conclusion, a prospective follow-up of a large cohort of recovered COVID-19 patients of varying degrees of severity and cared for in different settings suggested that nearly 50% of them had post-COVID-19 syndrome 6 months after onset. Several clinical and healthcare management issues should be considered in a long-term pandemic. In addition, the association of post-COVID-19 symptoms with serological response triggers more insights into the underlying pathogenetic mechanisms and possible therapeutic strategies.

Q10:  References should be corrected. Some of references are not clear (8, 9, 10, 11, 14, 15, 18,19,20 and other…)

 A10: We are sorry for our negligence. We corrected all the references.

Round 2

Reviewer 1 Report

The Authors revised the manuscript as recommended. However, it still contains minor editorial errors:
1. the city of residence of the reagent/programme manufacturer is not stated,
2. the tables are still not prepared in accordance with the Instructions for Authors, e.g. they contain vertical lines
3. figures should not be in frames

I ask the Authors to review the manuscript and correct it according to the Diagnostcs  guidelines

Author Response

Dear Reviewer 1,

Thank you very much for your review and constructive suggestions.

The Authors revised the manuscript as recommended. However, it still contains minor editorial errors:

Q1: the city of residence of the reagent/programme manufacturer is not stated,

A1: We added the residence of the program manufacturer.

Q2.:the tables are still not prepared in accordance with the Instructions for Authors, e.g. they contain vertical lines

A2:We changed the tables accordingly.

Q3:figures should not be in frames

A3: We tried to remove the frames, but that implies a decrease in image quality.

Reviewer 2 Report

Dear colleagues,

The following items should be corrected:

-          Figure 1, 2, 3 and 4 are not clear, it should be presented in one table with statistic analysis data and other characteristics of patients.  It is not results of the study. This data should be presented in the part Materials and Methods.

-          Table 1  are not clear.

All these concerns should be well addressed to consider this manuscript suitable for publication.

Author Response

Dear Reviewer 2,

Thank you very much for your review and constructive suggestions

Q1:  Figure 1, 2, 3 and 4 are not clear, it should be presented in one table with statistic analysis data and other characteristics of patients.  It is not results of the study. This data should be presented in the part Materials and Methods.

A1:One more time, we support that the figures are precise. They represent the results of the study (classifications of covid cases according to shape; distribution of cases according to the required number of hospital days).

Q2:   Table 1  are not clear.

A2: We don't understand precisely what you mean when you say it's not clear. n statistics, the Pearson correlation ― also known as Pearson's r, the Pearson product-moment correlation coefficient (PPMCC), the bivariate correlation, or the correlation coefficient ― is a measure of linear correlation between two sets of data. It is the ratio between the covariance of two variables and the product of their standard deviations; thus, it is a normalized measurement of the covariance, such that the result always has a value between −1 and 1.

A solid correlation was noted between patients' ages and the number of days of hospitalization, no matter their sex or background (Pearson's r correlation coefficient, p<0.0001). Therefore, older ages require more extended hospitalization, even with moderate or mild forms of the disease, mainly because of the persistence of specific symptomatology and more delayed negativity.

.